# Investigating the Effects of Nitric Acid Treatments on the Properties of Recycled Carbon Fiber

**DOI:** 10.3390/polym15040824

**Published:** 2023-02-07

**Authors:** Gyungha Kim, Hyunkyung Lee, Minsu Kim, Dae Up Kim

**Affiliations:** Carbon & Light Materials Application Group, Korea Institute of Industrial Technology, 222 Palbok-ro, Deokjin-gu, Jeonju 54853, Republic of Korea

**Keywords:** recycled carbon fiber, nitric acid, functional group, surface free energy, mechanism

## Abstract

In this study, the chemical state change of recycled carbon fiber (rCF) surfaces and the mechanism of the oxygen functional groups according to nitric acid treatment at various times and temperatures were investigated to upcycle the carbon fiber recovered from used carbon composite. When treated with nitric acid at 25 °C, the carbon fiber surface demonstrated the same tensile properties as untreated carbon fiber (CF) for up to 5 h, and the oxygen functional group and polar surface energy of C–O (hydroxyl group) and C=O (carbonyl group) increased slightly compared to the untreated CF up to 5 h. On the other hand, at 100 °C, the tensile properties slightly decreased compared to untreated CF up to 5 h, and the amount of C–O and C=O decreased and the amount of O=C–O (lactone group) started to increase until 1 h. After 1 h, the amount of C-O and C=O decreased significantly, and the amount of O=C–O increased rapidly. At 5 h, the amount of oxygen functional groups increased by 92%, and the polar surface energy increased by 200% compared to desized CF. It was determined that the interfacial bonding force increased the most because the oxygen functional group, O=C–O, increased greatly at 100 °C and 5 h.

## 1. Introduction

Carbon fiber (CF) is a lightweight material that has a high specific strength, stiffness, thermal conductivity, and electrical conductivity compared to other materials, as well as high corrosion resistance and chemical resistance [1,2,3,4,5]. However, since CF is an expensive material, it is applied only to expensive components such as those associated with aviation, space, wind power, sports cars, and sporting goods, and environmental problems such as disposal of carbon composites in landfills after use remain a challenge that must be solved [6]. To expand application to all industries, including automobiles, in the future, it is necessary to lower CF prices, regulate parts recycling, and upcycle technology for recovering and recycling waste carbon composites [7,8,9,10,11]. Currently, some research on how to recover recycled CFs from waste carbon composites is being conducted, but the recovered recycled CFs deteriorate by about 20% to 30% compared to recycled CFs, so their reuse is limited [9,12]. In particular, the recovered CF has a deteriorated surface, and the interfacial bonding strength between the CF and the resin decreases, so a surface treatment to improve the interfacial bonding strength is necessary to reuse the recovered CF [13,14].

In general, surface treatment includes wetting methods (liquid oxidation) such as sulfuric acid and nitric acid [14,15,16,17,18,19,20,21], dry methods (dry gaseous oxidation) [22,23,24,25,26,27,28,29] such as high-temperature treatment in an atmosphere of oxidizing gas or inert gas, electric oxidation methods [3,30,31,32], the plasma method [33,34,35,36,37,38,39] of surface treatment using ionized gas, and exposure to strong energy such as ozone and ultraviolet rays (energetic ions oxidation) [40,41,42].

Surface treatment changes physical/chemical properties depending on the type of treatment solution/gas, treatment temperature, and treatment time. As a result of treatment with sulfuric acid/nitric acid at a temperature between 25 °C and 60 °C, the surface damage and the amount of O=C–O (lactone group) and C=O (carbonyl group) in oxygen functional groups significantly increases as the treatment temperature and the amount of sulfuric acid increase [20]. As a result of analyzing the physical/chemical properties of CF according to the heat treatment temperature in a vacuum atmosphere, it was found that as the heat treatment temperature increased, the amount of C–O (hydroxyl group) in CF decreased, increasing the contact angle, decreasing the polar surface free energy, and decreasing the IFSS [26]. As a result of anodic oxidation surface treatment using sodium hydroxide and sulfuric acid, the higher the concentration of acid, the more pores. Additionally, the damaged area and the increase of COOH (carboxyl group) and C–O introduced to the surface of CF was large [30]. As a result of plasma treatment in an argon atmosphere, it has been reported that the longer the treatment time, the greater the damage to the CF and the lower the tensile strength, but the C–O decreases and the COOH increases, resulting in an increase in the interfacial bonding force between the CF and the resin [37].

In this study, nitric acid treatment was introduced as a surface treatment process for upcycling regenerated CF, and optimal conditions for improving the interfacial bonding force between CF and resin were derived through mechanical and chemical property analysis according to treatment temperature and time. In addition, the chemical state changes of carbon, oxygen, nitrogen, and silicon present on the surface of carbon fiber according to the nitric acid treatment temperature and time and the oxygen functional group mechanism were identified.

## 2. Experimental Details

The carbon fiber in this research is recycled carbon fiber recovered from used carbon composite. Table 1 compares the properties of commercial CF (untreated CF) from Toray (Tokyo, Japan) and the recycled carbon fiber (recycled CF) used in this research. Carbon fibers of various qualities exist depending on the manufacturer and manufacturing process of carbon fibers. In this study, the characteristics of rCF were obtained using carbon fibers, as shown in Table 1, and rCF deteriorated and showed about 20% lower characteristics compared to untreated CF, a result proven by other researchers [9,12].

The CF used in the experiment was desized and then surface treated with nitric acid. Nitric acid treatment was performed under a temperature of 25 °C to 100 °C and a time of 0.5 to 5 h. Surface changes of the CF treated with nitric acid were observed under an acceleration voltage of 20 kV using FE–SEM. To evaluate the mechanical properties of CF, a single fiber tensile test was conducted under a tensile speed of 5 mm/min according to the ASTM D3822 standard, and the average values from testing 25 times for each test condition were used. X-ray photoelectron spectroscopy (XPS) of the Nexsa XPS system (Jeonju-si, Republic of Korea, Korea Basic Science Institute) was used to analyze the change in chemical functional groups on the surface of the CFs according to acid treatment conditions. The specimen was irradiated with monochromatic Al Kα (1486.6 eV), and high-resolution spectra were obtained with a 400 μm sized beam and a pass energy of 50 eV. Dynamic contact angle measurement is required to analyze changes in surface energy. In this study, hydrophilic water and hydrophobic diiodomethane were used to measure the dynamic contact angle, and the dynamic contact angle and surface free energy were calculated from the values measured according to the Wilhelmy plate method for each condition. The value was calculated after a second evaluation.

## 3. Results and Discussion

### 3.1. Surface Topography of Carbon Fibers

Figure 1 shows the results for comparing and observing the surface of the CF after nitric acid treatments at various time durations and temperatures. When treated with nitric acid at 25 °C to 100 °C for 1 to 5 h, the changes in the surface of the CF could not be observed. Although we increased the amplification for SEM photographs of carbon fibers, there was no change. According to another researcher, surface treatment with sulfuric acid/nitric acid removed sizing for up to 15 min at 60 °C, resulting in vertical stripes though the surface was not damaged; after 15 min, the surface was grooved and damaged [20]. It has been reported that there was almost no surface reaction as a result of heat treatment at 600 °C~700 °C in H_2_/Ar atmosphere [27] (an inert gas) and that a defect exists on the surface of a CF at 600 °C or higher in cases of heat treatment in a nitrogen atmosphere [28]. In addition, in the case of plasma treatment in an oxygen atmosphere, the surface of the CF was seriously damaged after 5 min, the diameter of the fiber was rapidly reduced, and some of it was lost at 7 min, resulting in loss of the function of the CF [38]. This is believed to increase the number and size of pores that exist due to chemical reactions on the surface of CF, depending on the degree of exposure to liquid and gas to be treated during surface treatment. In addition, when treated with nitric acid at 100 °C for 5 h—the experimental condition of this study—no change in the CF surface was observed.

### 3.2. Tensile Properties of Carbon Fibers

The tensile properties according to the temperature and treatment time of nitric acid were evaluated, and the results are shown in Figure 2. The tensile strength, modulus, and elongation after nitric acid treatment demonstrated similar characteristics. In other words, it was confirmed that the tensile properties of nitric acid at 25 °C for 5 h and at 100 °C for 3 h were almost similar to untreated CF but slightly decreased compared to untreated CF after 3 h at 100 °C. Changes in tensile strength between 25 °C and 100 °C were similar to changes at 25 °C within the margin of error. Ibarra et al. showed that when surfaces were treated with nitric acid at 80 °C, tensile strength had no significant difference from untreated CF up to 7 h but that it decreased to 80% compared to untreated CF at 12 h [19]. Rong et al. reported that when heat treatment was performed in an oxygen atmosphere, pitting occurred on the surface of the fiber, and the surface area increased without a change in tensile strength at 420 °C for up to 1 h, but after 2 h, the tensile strength decreased, the pitting agglomerated, and the surface area gradually decreased [29]. Lee et al. showed that plasma treatment in an oxygen atmosphere was almost similar to untreated CF for up to 1 min but showed a rapid decrease up to approximately 52% compared to untreated CF at 5 min [38]. In general, when CF is surface treated, as the temperature, treatment time, and treatment energy increase, surface erosion and carbon and oxygen inside the CF react violently, resulting in the deterioration of the carbon fiber and loss of mechanical strength. It is considered that the effect of temperature on the change of CF is greater than the effect of time during surface treatment with nitric acid.

### 3.3. Surface Composition of Carbon Fibers

Figure 3 is the XPS spectra for analyzing the chemical changes on the CF surface according to the nitric acid treatment conditions, and the composition changes and O/C ratios are summarized in Table 2. When the surface treatment with nitric acid was at 25 °C, as the treatment time increased from 1 to 5 h, the amount of carbon decreased, and the amount of oxygen, nitrogen, and silicon increased significantly compared to the desized CF. By contrast, regarding the composition change before and after the surface treatment at 100 °C with nitric acid, as the treatment time increased to 5 h, the amount of carbon decreased, the amount of oxygen and nitric acid increased, and the amount of silicon significantly decreased compared to the desized CF. The increase in oxygen functional groups between 25 °C and 100 °C was smaller than the amount at 100 °C and larger than that at 25 °C.

For the O/C ratio, which is an indicator of interfacial shear strength, as the temperature and time increased from 25 °C to 100 °C, from 1 to 5 h, it increased to 0.33 compared to untreated CF at 100 °C and 5 h [17]. Observation of the C1s spectra revealed that after desizing, the amount of C–O (hydroxyl groups) and C=O (carbonyl groups) present in large amounts on the surface of the CF was significantly reduced. Furthermore, the amount of C–O and C=O when the surface was treated in nitric acid at 25 °C until 1 h decreased slightly compared to the desized CF, and as it increased from 1 to 5 h, the amount of C–O and C=O increased slightly compared to 1 h, and O=C–O (lactone group) was newly created (Figure 3). In addition, as the time increased from 1 to 5 h, the bond between the carbon fiber surface and nitrogen present in nitric acid rapidly expanded, resulting in significant increases in the amount of C–N compared to the desized CF. The amount of silicon present inside the CF also increased due to the exposed surface. However, as the surface treatment in nitric acid increased from 1 to 5 h at 100 °C, the amount of C–O and C=O decreased slightly compared to the desized CF. Moreover, O=C–O was formed and expanded rapidly, and the amount of C–N increased up to 1 h at 100 °C. The amount of C–N expanded due to the reaction between carbon present on the CF surface and nitrogen of nitric acid, but after 1 h, the amount of C–N decreased slightly compared to 1 h due to the combination and removal of oxygen present on the CF and nitrogen of nitric acid. In addition, the silicon present inside the CF was removed through a violent reaction on the surface of the CF as the time increased from 1 h to 5 h, and the amount of silicon decreased compared to the desized CF. This is because the carbon present on the surface of the CF exposed by desizing during surface treatment in nitric acid at 25 °C combines with the oxygen of nitric acid and is removed as CO_2_, so the carbon inside the CF is slightly reduced, and the silicon inside the CF is revealed, revealing the amount of silicon. It is thought that C–O, C=O, and C–N increased and O=C–O was produced because the oxygen and nitrogen in nitric acid were bound to the CF surface. On the other hand, when the surface was treated with nitric acid at 100 °C, a strong bond was created between the carbon present on the surface of CF and the oxygen present in the nitric acid, and oxygen was introduced into the C–O and C=O bonds, resulting in the formation and rapid increase of O=C–O bonds. It is believed that silicon was rapidly reduced by the removal of carbon fiber due to the strong energy on the surface of the CF.

To see the change of the peak according to the composition of the functional group formed on the surface of the CF, C1s were separated and shown in Figure 4 and Table 3. As can be seen from the above results, when it increased from 1 to 5 h at 25 °C of nitric acid, C–O, C=O, C–N, and O=C–O slightly increased compared to 1 h. On the other hand, at 100 °C, C–O and C=O decreased sharply compared to untreated CF, and O=C–O increased significantly. After surface treatment with nitric acid, the reaction between CF and nitric acid intensified as the temperature increased from 25 °C to 100 °C, and treatment time increased from 1 to 5 h, resulting in increases in O=C–O, which greatly affected the interfacial shear strength with resin in oxygen functional groups.

In previous studies, as a result of immersing sulfuric acid/nitric acid at a ratio of 3:1 at 60 °C for 15 min, oxygen increased 16 times compared to untreated CF. Furthermore, nitrogen was produced and increased 1.3 times, and Si was augmented by 1.0 times. The amount of oxygen most likely increased due to the increase in carboxyl groups after surface treatment, and it has been reported that nitrogen and Si was produced by oxidized nitrogen and sulfur [20]. In addition, it has been shown that oxygen present in the atmosphere combined with carbon on the CF surface during heat treatment in an oxygen atmosphere, resulting in N–H being broken and the amount of N–O, C–O, and C=O being augmented by combining with oxygen [23]. After heat treatment in the atmosphere of H_2_/Ar, it was reported that the surface change was slight by hydrogen, and there was almost no chemical change [27]. In the case of plasma treatment in an argon atmosphere, it has been reported that no new oxygen functional group was generated on the surface of the CF, which has only the effect of desizing because an inert gas was used [39]. In this study, as the temperature and time increased during nitric acid treatment, the oxygen functional groups introduced on the CF surface increased, and the defects on the CF surface augmented, which is expected to increase interfacial shear strength.

### 3.4. Surface Energy Analysis

To examine the changes in the surface free energy of CFs after nitric acid treatment, the dynamic contact angle was measured using a hydrophilic and hydrophobic wetting solution, and the measured values were substituted into the following equation to obtain polar surface free energy and nonpolar surface free energy values:(1)$$\frac{\gamma_{L}(1+\cos\theta )}{2{\left(\gamma_{L}^{D}\right)}^{\frac{1}{2}}}={\left(\gamma_{S}^{P}\right)}^{\frac{1}{2}}\times {\left(\frac{\gamma_{L}^{P}}{\gamma_{L}^{D}}\right)}^{\frac{1}{2}}+{\left(\gamma_{S}^{D}\right)}^{\frac{1}{2}}$$
where $\gamma_{L}$ is the total surface energy of the wetting liquid, $\gamma_{L}^{D}$ is the nonpolar surface energy of the wetting liquid, $\gamma_{L}^{P}$ is the polar surface energy of the wetting liquid, $\gamma_{S}$ is the total surface energy of the specimen, $\gamma_{S}^{D}$ is the nonpolar surface energy of the specimen, $\gamma_{S}^{P}$ is the polar surface energy of the specimen, and θ is the contact angle. The polar surface energy of the specimen (i.e., the slope) and the nonpolar surface energy (i.e., the Y-intercept) were obtained from the two coordinates calculated by substituting the advance angle, which is the angle when the specimen enters the hydrophilic or hydrophobic wetting liquid, into Equation (1). The change in the dynamic contact angle of the CF according to the temperature and treatment time with nitric acid is shown in Figure 5. At 25 °C in nitric acid, the contact angle decreased compared to untreated CF as the treatment time increased from 30 min to 5 h. At 100 °C in nitric acid, the contact angle decreased compared to untreated CF from 30 min to 1 h, but after 1 h, the contact angle gradually increased compared to 1 h.

Figure 6 is the result of summarizing the surface free energy and polar free energy according to the temperature and time of the nitric acid treatment of the dynamic contact angle results. At 25 °C, the surface free energy and polar free energy of nitric acid gradually increased as time increased from 1 to 5 h. At 100 °C and for up to 1 h, the surface energy and polar free energy of nitric acid increased compared to 0.5 h, and then after 1 h, the surface energy and polar free energy decreased compared to 1 h. It is judged that the contact angle decreases and the polar surface energy significantly increases due to the active introduction of oxygen functional groups at 100 °C for 1 h, and after 1 h, the CF surface erodes, expanding the contact angle and reducing the polar surface energy. The change in surface energy between 25 °C and 100 °C was larger than that at 25 °C and smaller than the change at 100 °C, and the value of polar surface energy was greater than the value of maximum polar surface energy at 25 °C, but it was smaller than the value of the maximum polar surface energy at 100 degrees.

Looking at the polar/surface free energy ratio in Figure 7, the effect of nitric acid treatment time was small, but the effect of temperature was large. That is, at 25 °C, the polar/surface free energy ratio slightly increased as the time increased from 30 min to 5 h, and at 100 °C, the ratio increased from 30 min to 1 h but decreased after 1 h. From this, it was determined that the oxygen functional group was greatly increased at 100 °C for 1 h in nitric acid, resulting in a decrease in contact angle and a significant advancement in polar surface energy.

According to one researcher, when the surface is treated with nitric acid at 80 °C for 4 h, oxygen functional groups are introduced 17 times, the contact angle of CF is reduced, and the surface energy is increased, thereby amplifying the interfacial bonding force between CF and resin [14]. In the case of heat treatment at 300 °C in a nitrogen atmosphere, it has been reported that the polar free energy decreases due to the reduction of oxygen functional groups resulting from the curing of sizing [26]. In addition, after the electrical oxidation treatment, it was confirmed that the polar free energy increased due to the increase in the surface area and the increase of C–O, C=O, and O=C–O by etching the surface of the CF [3]. From these results, it can be seen that the interfacial shear strength between the CF and the resin increases due to the increase in polar free energy resulting from the introduction of oxygen functional groups to the surface of the CF during surface treatment.

### 3.5. Functional Group Change Mechanism by Nitric Acid Treatment

Based on the analysis of the mechanical/chemical properties of CFs according to the temperature and treatment time of nitric acid treatment, the surface change and oxygen functional group mechanism of CFs are schematized, as shown in Figure 8. When desizing with acetone, the sizing is removed and the oxygen atoms of C–O (hydroxyl group) and C=O (carbonyl group) present on the surface of the CF and the oxygen atoms in acetone are combined, as shown in Equation (2), to be removed as O_2_ and CO_2_. As the treatment with nitric acid increased from 1 to 5 h at 25 °C, C–C and C=C on the surface of the CF were continuously bonded with the oxygen of the nitric acid (see Equation (2)) to remove O_2_ and CO_2_, and oxygen was bonded to the exposed surface, which generated and increased C–O and C=O, which were removed during desizing, creating O=C–O (lactone group) newly. In addition, carbon present on the surface of CF and nitrogen in nitric acid were combined to increase C–N, and silicon present inside the CF was revealed, and the amount of silicon increased. On the other hand, at 100 °C, the CF surface was revealed as it increased from 1 to 5 h due to strong energy, and the binding of C–C and C=C was significantly reduced. The oxygen of nitric acid was strongly bonded to the C–O and C=O present on the exposed CF surface to generate O=C–O and rapidly increase O=C–O, and Si decreased as carbon fibers were damaged by nitric acid and Si was removed. In this study, it is judged that O=C–O increases for more than 5 h at 100 °C in nitric acid, but the carbon inside the CF and the oxygen atoms of nitric acid reacted violently, and the CF properties deteriorated rapidly.
(2)$$\begin{array}{c} C-O+O\;\\ C=O+O\; \end{array}\to \;C+O_{2}↑+{\;\mathrm{CO}}_{2}↑$$

## 4. Conclusions

In this study, the mechanical and chemical properties of CFs according to temperature and treatment time with nitric acid were analyzed, and the chemical state change and oxygen functional group mechanism of the CF surface according to the nitric acid treatment conditions were observed. There was no significant difference in tensile properties from untreated CF at 25 °C from 1 to 5 h, and the amount of C–C, C–O, and C=O on the carbon fiber surface was combined with oxygen in nitric acid to reduce C–C and C=C, and oxygen was introduced into C–O and C=O to form O=C–O for up to 5 h. In addition, the carbon present on the surface of the CF and the oxygen of nitric acid combine to increase C–N. Additionally, at 100 °C in nitric acid, the tensile properties do not differ significantly from the untreated CF for up to 1 h. However, after 1 h, tensile properties gradually decrease, C–O and C=O react violently in heated nitric acid, and O=C–O is rapidly produced and expands. Furthermore, the oxygen functional group is greatly increased and decreased by combining Si on the exposed surface with oxygen. As a result of this, the polar free energy increased significantly at 100 °C compared to 25 °C and showed the most optimal contact angle and polar free energy at 100 °C for 1 h. Therefore, within the scope of this study, after surface treatment of nitric acid at 100 °C for 1 h, the most optimal oxygen functional group, polar surface energy, and interfacial bonding force were exhibited without a decrease in tensile strength.

Future experiments should focus more on evaluating whether the mechanical characteristics of composites manufactured using surface-treated CF with nitric acid under optimal conditions are equivalent to commercial carbon composites. This can contribute to the commercialization of automobile parts manufactured using this method.

## Figures and Tables

**Figure 1 polymers-15-00824-f001:**
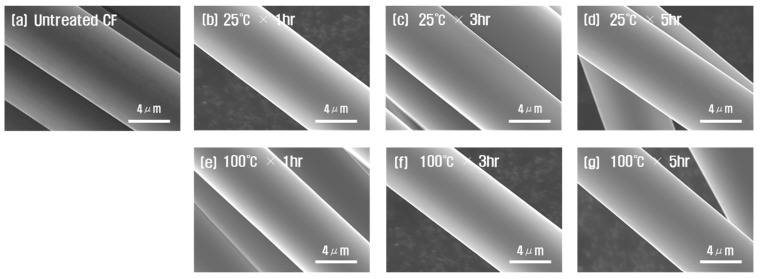
SEM photographs of carbon fibers according to chemical treatment conditions: (**a**) untreated CF, (**b**) 1 h, (**c**) 3 h, (**d**) 5 h at 25 °C and (**e**) 1 h, (**f**) 3 h, (**g**) 5 h at 100 °C.

**Figure 2 polymers-15-00824-f002:**
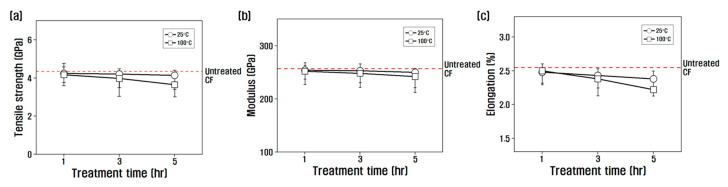
Effect of chemical treatment according to temperature and treatment time on the tensile properties of carbon fiber (**a**) tensile strength and (**b**) modulus (**c**) elongation.

**Figure 3 polymers-15-00824-f003:**
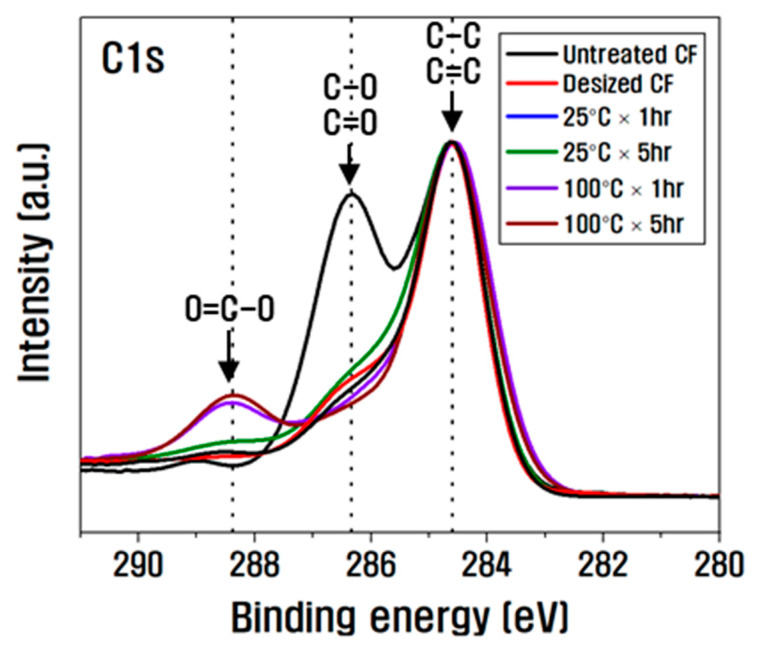
C1s XPS spectra of chemical treatment according to temperature and treatment time.

**Figure 4 polymers-15-00824-f004:**
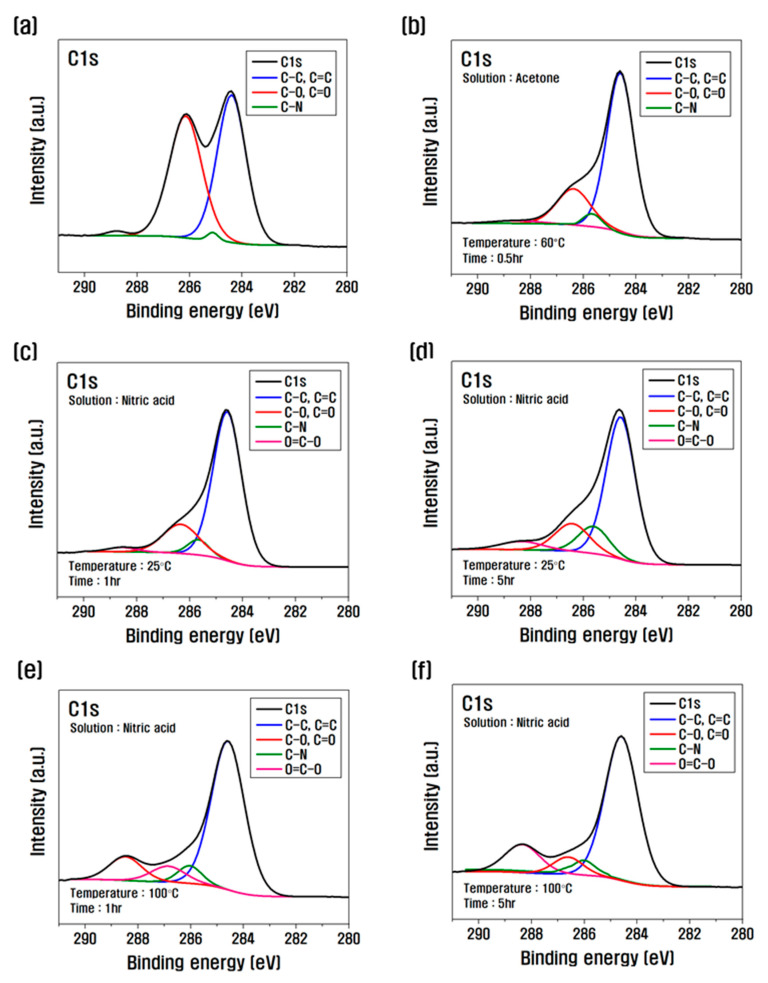
XPS photoelectron spectra of C1s after chemical treatment: (**a**) untreated CF, (**b**) desized CF (**c**) 1 h, (**d**) 5 h at 25 °C and (**e**) 1 h, (**f**) 5 h at 100 °C.

**Figure 5 polymers-15-00824-f005:**
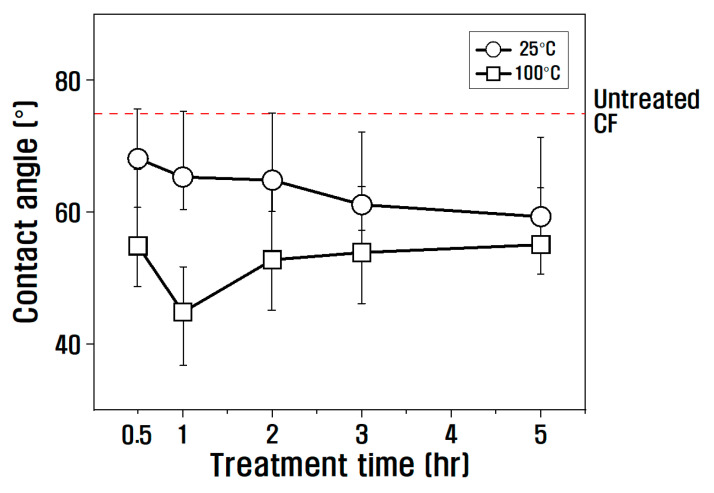
Relationship between contact angle and treatment time according to temperature.

**Figure 6 polymers-15-00824-f006:**
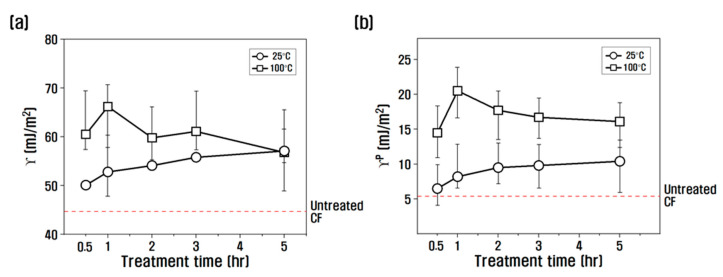
Variation of (**a**) surface free energy and (**b**) polar surface energy of chemical treatment on carbon fibers.

**Figure 7 polymers-15-00824-f007:**
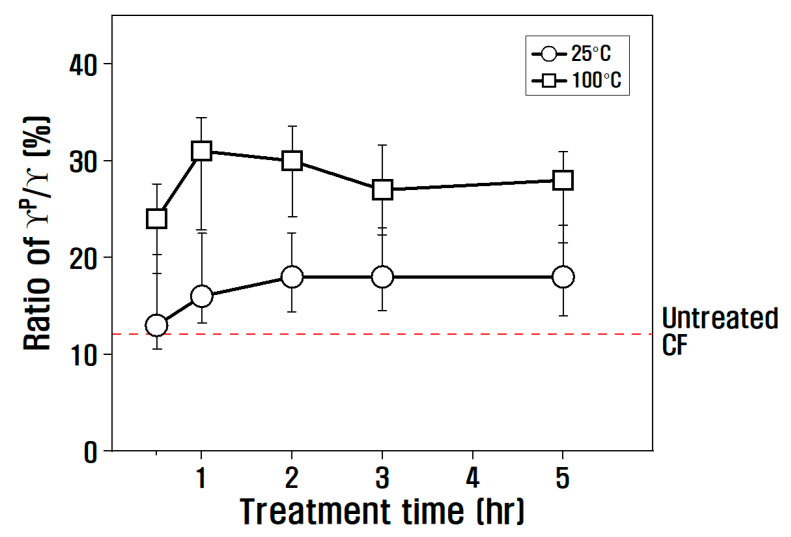
Variation of $\gamma^{P}/\gamma$ according to chemical treatment conditions.

**Figure 8 polymers-15-00824-f008:**
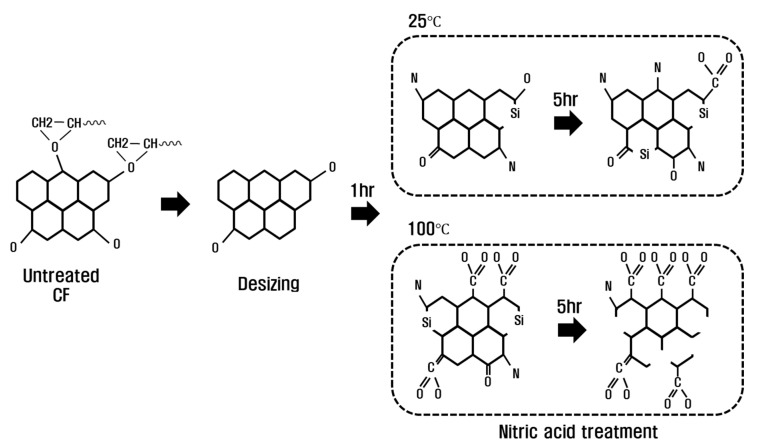
Schematic of the chemical reaction of carbon fiber according to chemical treatment temperature and treatment time.

**Table 1 polymers-15-00824-t001:** Properties of carbon fiber in this study.

Type	Untreated CF	Recycled CF
Tensile strength (Gpa)	4.49	3.45
Modulus (Gpa)	261	256
Elongation (%)	2.62	2.08
Density (g/cm^3^)	1.80	1.80

**Table 2 polymers-15-00824-t002:** Effect of chemical treatment conditions on the surface elemental composition of carbon fibers.

Treatment Condition	Elemental Composition (at.%)				O/C
	Carbon	Oxygen	Nitrogen	Silicon	
Untreated CF	76.31	21.31	0.75	1.63	0.28
Desized CF	83.83	12.46	1.88	1.83	0.15
25 °C × 1 h	76.23	17.76	3.16	2.85	0.23
25 °C × 5 h	72.59	20.39	3.65	3.37	0.28
100 °C × 1 h	73.91	22.81	2.73	0.55	0.31
100 °C × 5 h	73.03	24.03	2.70	0.24	0.33

**Table 3 polymers-15-00824-t003:** Functional group according to temperature and treatment time by XPS.

Treatment Condition	C1s (at.%)			
	C–C, C=C	C–O, C=O	C–N	O=C–O
Untreated CF	71.09	26.86	0.98	1.07
Desized CF	75.59	20.10	2.21	2.10
25 °C × 1 h	74.07	18.06	5.66	2.20
25 °C × 5 h	66.15	20.11	8.14	5.60
100 °C × 1 h	74.00	8.12	6.28	11.59
100 °C × 5 h	70.59	8.00	6.05	15.36

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
