# Peer review of "Investigating the Effects of Nitric Acid Treatments on the Properties of Recycled Carbon Fiber"

_polymers, 2023, doi:10.3390/polym15040824_

Round 1

Reviewer 1 Report

The authors study changes in the physical and chemical properties of carbon fibers caused by various kinds of surface treatment.  The paper is interesting and provides helpful information for scientists working on carbon fibers.  I think the paper can in principle be published as is.  I have only one minor comment:  Fig.1 could be removed.  As the authors say, no changes on the surface can be seen.  Thus, mentioning this fact in the text should be sufficient. On the other hand, in Fig. 1 only an extremely small area of a few square micron is shown.  Can the authors add a brief statement on how they determined that no changes on the entire surface of all carbon fibers took place.

Author Response

Point 1: Fig.1 could be removed.  As the authors say, no changes on the surface can be seen.  Thus, mentioning this fact in the text should be sufficient. On the other hand, in Fig. 1 only an extremely small area of a few square micron is shown.  Can the authors add a brief statement on how they determined that no changes on the entire surface of all carbon fibers took place.

Response 1: There is no change in SEM within the scope of this study, but I think there should be Figure 1 because data proving it is needed. However, as you advised, it is difficult to prove no change at this distance at 4 micron amplification. To prove this, we added an explanation that there is no change even if the amplification is increased.

Reviewer 2 Report

The authors carried out an experiemental investigation about the surface treatment (nitric acid)in recycled carbon fiber. The topic is very interesting but the authors should make an effort in order to describe the priorities of this work and the goals.

I would like to make somme sugestions:

1. The recycled carbon fiber should be mention in the abstract or even in the title. It appears within the experimental details.

2.The abstarct is a summary of the experimental work and it is bague "It is believed,.. from 25 C-100C and from 1-5 h,.... resulting tin greater interfacial bonding force between carbon and resin".

3. I don´t know if this behaviour can be applied only to the CF of table 1

4.It is convenient to have data related to an intermediate temperature between 25 C (room temperature) and 100ºC.

5 can you define the optimal conditions to be applied to the anitric acid treatments?

Author Response

Thank you for giving us this opportunity to submit a revised draft of our paper entitled "Investigating the effects of nitric acid treatments on the properties of carbon fiber”. We appreciate the time and effort the reviewers have dedicated to providing valuable feedback on our manuscript. We have revised the manuscript according to their feedback, and the answers to the reviewer's comments follow below

We look forward to responding to any further questions and comments you may have.

------------------------------------------------------------------------------------------------------------------------------------

Point 1: 1. The recycled carbon fiber should be mention in the abstract or even in the title. It appears within the experimental details.

Response 1: We agree with the reviewer’s comment, and we mentioned recycled carbon fiber in the title and abstract.

Point 2: The abstarct is a summary of the experimental work and it is bague "It is believed,.. from 25°C -100°C and from 1-5 h,.... resulting tin greater interfacial bonding force between carbon and resin".

Response 2:  Following the reviewer’s advice, we have removed the ambiguity. Therefore, I modified it to the following sentence.

“It was determined that the interfacial bonding force increased the most because the oxygen functional group, O=C-O, increased greatly at 100°C and 5 hours.”

Point 3: I don´t know if this behaviour can be applied only to the CF of table 1

Response 3: As reviewer’s mentioned, the values in Table1 may not apply to other carbon fibers. However, as other researchers have proven that recycled carbon fiber deteriorates by about 20-30% compared to commercialized carbon fiber[9,12], the values of the physical properties of recycled carbon fiber similar to those in this paper can be confirmed.

Point 4: It is convenient to have data related to an intermediate temperature between 25°C (room temperature) and 100°C.

Response 4: As reviewer’s mentioned, we have done all the tests at temperatures between 25 and 100°C. Since the values of tensile strength, oxygen functional groups, and surface energies at that temperature were between 25 and 100°C, they were not indicated in the thesis.

Point 5: can you define the optimal conditions to be applied to the anitric acid treatments?

Response 5: Our research has defined the optimal conditions for nitric acid treatment within the experimental range. In this study, the optimal condition was defined as the highest value of oxygen functional group and interfacial bonding force without deterioration of tensile strength after nitric acid treatment. Therefore, after surface treatment of nitric acid at 100°C for 1 hour, the most optimal oxygen functional group, polar surface energy, and interfacial bonding force were exhibited without a decrease in tensile strength.

Reviewer 3 Report

The manuscript included investigation of surface treatment of carbon fibers (CFs) with nitric acid at varying time and temperature. The authors were thorough in their design of the experiment and presented the observed data in a meaningful way. However, I believe this manuscript should be improved in order to get it published. I recommend the following suggestions to fix certain errors and improve the overall quality of the manuscript.

On the 2nd paragraph of page 2, it is stated "In addition, the chemical state change and oxygen functional group mechanism of the carbon fiber surface according to the nitric acid treatment temperature and time were identified". I am not sure what the authors meant by chemical state change. I will suggest elaborating a little bit and reconstruct this sentence.

On page 2, paragraph 4, authors mentioned that they used "diodemethane" to measure the dynamic contact angle. I believe that there is a typo with the chemical name as I failed to find the existence of such a chemical name.

The SEM images presented in the manuscript to exhibit no apparent changes on the surface are taken at 4 micron amplification and it is hard to see any changes from this distance. I will suggest increasing the amplification if they wat to prove that there are no surface changes.

Lastly, inclusion of FT-IR characterization of the treated CFs will surely help to identify the functional groups authors are claiming to incorporate on the surface. I will suggest getting such characterizations to be done and included in the manuscript.

The chemical equations proposed on page 10 suggested SiO2 to be a gaseous product (equation 2), which requires the reactions to be held at extremely high temperature. This is needed to be corrected.

If the issues are fixed or properly addressed, this manuscript can be published.

Author Response

Thank you for giving us this opportunity to submit a revised draft of our paper entitled "Investigating the effects of nitric acid treatments on the properties of carbon fiber”. We appreciate the time and effort the reviewers have dedicated to providing valuable feedback on our manuscript. We have revised the manuscript according to their feedback, and the answers to the reviewer's comments follow below

We look forward to responding to any further questions and comments you may have.

------------------------------------------------------------------------------------------------------------------------------------

Point 1: On the 2nd paragraph of page 2, it is stated "In addition, the chemical state change and oxygen functional group mechanism of the carbon fiber surface according to the nitric acid treatment temperature and time were identified". I am not sure what the authors meant by chemical state change. I will suggest elaborating a little bit and reconstruct this sentence.

Response 1: We agree with the reviewer’s comment, and we modified the sentence in the 2nd paragraph of page 2 to clarify the following:

“In addition, the chemical state change of carbon, oxygen, nitrogen, and silicon present on the surface of carbon fiber according to the nitric acid treatment temperature and time and the oxygen functional group mechanism were identified.”

Point 2: On page 2, paragraph 4, authors mentioned that they used "diodemethane" to measure the dynamic contact angle. I believe that there is a typo with the chemical name as I failed to find the existence of such a chemical name.

Response 2: Thank you for finding the error in this paper. As reviewer’s mentioned, we corrected the misspelling in the paper as follows: “diiodomethane”

Point 3: The SEM images presented in the manuscript to exhibit no apparent changes on the surface are taken at 4 micron amplification and it is hard to see any changes from this distance. I will suggest increasing the amplification if they wat to prove that there are no surface changes.

Response 3: As reviewer’s advised, it is difficult to prove no change at this distance at 4 micron amplification. To prove this, we added an explanation to the paper that increasing the amplification did not change anything.

Point 4: Lastly, inclusion of FT-IR characterization of the treated CFs will surely help to identify the functional groups authors are claiming to incorporate on the surface. I will suggest getting such characterizations to be done and included in the manuscript.

Response 4: As the reviewer’s advice suggests, FT-IR will be helpful in analyzing chemical composition changes. However, in this study, even the results from XPS were able to sufficiently confirm the change of oxygen functional group, and the surface energy and polar surface energy prove the cause of the change of oxygen functional group, so the analysis result of this study is considered to be sufficient. In the future, I will add FT-IF to make it a better study.

Point 5: The chemical equations proposed on page 10 suggested SiO2 to be a gaseous product (equation 2), which requires the reactions to be held at extremely high temperature. This is needed to be corrected.

Response 5: Following reviewer’s advice, the reaction of SiO2 at 100°C does not seem to occur, so the equation of SiO2 is removed. Therefore, the cause of the decrease in Si was identified only as damage to the carbon fiber by nitric acid, and the corresponding sentence was modified.

Round 2

Reviewer 2 Report

The answers giben in points 3,4 and 5 should be incorporated in this paper. Also, he novelty of this work should be underline

Author Response

Thank you for giving us this opportunity to submit a revised draft of our paper entitled "Investigating the effects of nitric acid treatments on the properties of carbon fiber”. We appreciate the time and effort the reviewers have dedicated to providing valuable feedback on our manuscript. We have revised the manuscript according to their feedback, and the answers to the reviewer's comments follow below

We look forward to responding to any further questions and comments you may have.

------------------------------------------------------------------------------------------------------------------------------------

Point 3: I don´t know if this behaviour can be applied only to the CF of table 1

Response 3: As reviewer’s mentioned, the paper states:

In chapter 2 of the paper, it was described as follows:

Carbon fibers of various qualities exist depending on the manufacturer and manufacturing process of carbon fibers. In this study, the characteristics of rCF were obtained using carbon fibers as shown in Table 1, rCF deteriorated and shows about 20% lower characteristics compared to untreated CF, a result proven by other researchers[9.12].

Point 4: It is convenient to have data related to an intermediate temperature between 25°C (room temperature) and 100°C.

Response 4: As the reviewer noted, the paper states:

In chapter 3.2. Tensile properties of carbon fibers of the paper, it was described as follows:

Changes in tensile strength between 25℃ and 100℃ were similar to changes at 25℃ within the margin of error.

In chapter 3.3. Surface composition of carbon fibers of the paper, it was described as follows:

The increase in oxygen functional groups between 25 °C and 100 °C was smaller than the amount at 100°C and larger than that at 25°C.

In chapter 3.4. Surface energy analysis of the paper, it was described as follows:

The change in surface energy between 25°C and 100°C was larger than that at 25°C and smaller than the change at 100°C, and the value of polar surface energy was greater than the value of maximum polar surface energy at 25°C, but it was smaller than the value of the maximum polar surface energy at 100 degrees.

Point 5: can you define the optimal conditions to be applied to the anitric acid treatments?

Response 5: As the reviewer noted, the paper states:

In conclusion of the paper, it was described as follows:

Therefore, within the scope of this study, after surface treatment of nitric acid at 100°C for 1 hour, the most optimal oxygen functional group, polar surface energy, and interfacial bonding force were exhibited without a decrease in tensile strength.

Reviewer 3 Report

I believe the authors have responded well to the suggested changes and I am satisfied with the responses. I recommend the manuscript to be published in the current format.

Author Response

We appreciate the time and effort the reviewers have dedicated to providing valuable feedback on our manuscript.